# PSR: PAIRWISE SUBJECT-CONSISTENCY REWARDS FOR MULTI-SUBJECT PERSONALIZED IMAGE GENERATION

## ABSTRACT

Personalized generation models for a single subject have demonstrated remarkable effectiveness, highlighting their significant potential. However, when extended to multiple subjects, existing models often exhibit degraded performance, particularly in maintaining subject consistency and adhering to textual prompt. We attribute these limitations to the absence of high-quality multi-subject datasets and the lack of refined post-training strategies. To address these challenges, we construct a scalable multi-subject data generation pipeline, which leverages strong single-subject models to synthesize multi-subject training data. Using this dataset, we first enable single-subject personalization models to acquire knowledge of multi-image and multi-subject scenarios. Furthermore, to enhance both subject consistency and text controllability, we design a set of pairwise subject-consistency rewards and general-purpose rewards, which are incorporated into a refined reinforcement learning stage. To comprehensively evaluate multi-subject personalization, we introduce a new benchmark that assesses model performance using seven subsets across three dimensions. Extensive experiments demonstrate the effectiveness of our approach in advancing multi-subject personalized image generation.

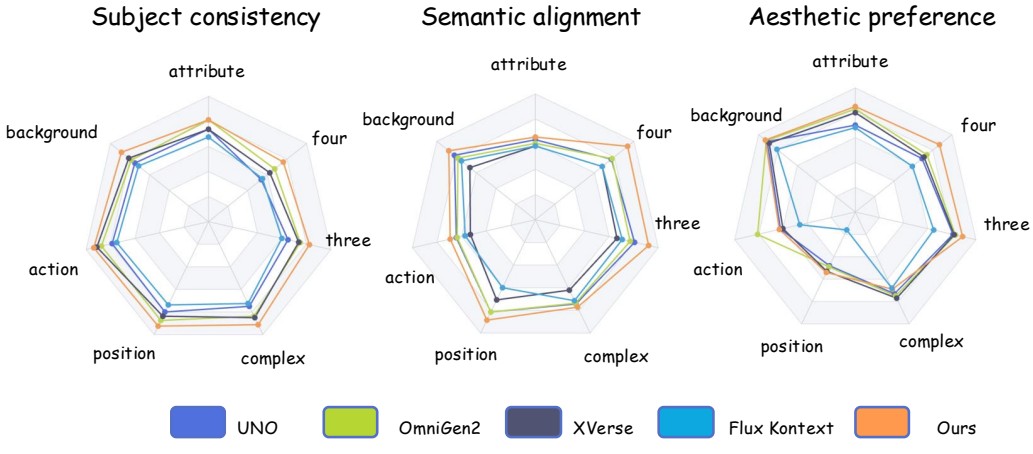

Figure 1: Quantitative analysis of existing methods on PSRBench.

## 1 INTRODUCTION

Personalized image generation aims to produce images that remain faithful to the given subjects while following textual instructions, and has significant applications in film production, personalized marketing, and beyond. Single-subject personalization models, such as Flux Kontext (Labs et al., 2025), have already demonstrated impressive capabilities. Meanwhile, several recent efforts,

including UNO (Wu et al., 2025b) and OmniGen (Xiao et al., 2025; Wu et al., 2025a), have begun to explore the domain of multi-subject generation, enabling models to accept multiple reference images and roughly maintain the overall subject identity. However, these multi-subject personalization approaches still suffer from several limitations: (1) Poor subject consistency – the subjects in the generated images may not similar to the given reference subjects, or even omit certain subjects entirely; (2) Limited adherence to text prompts – for example, given the prompt "the dog wears a chef's hat, and the cat wears a scarf". Eisting models may fail to capture specific attributes, such as a chef's hat, or incorrectly assign them, For instance, the generated image may depict the dog wearing a scarf and the cat wearing a chef's hat, failing to follow the semantics specified in the prompt.

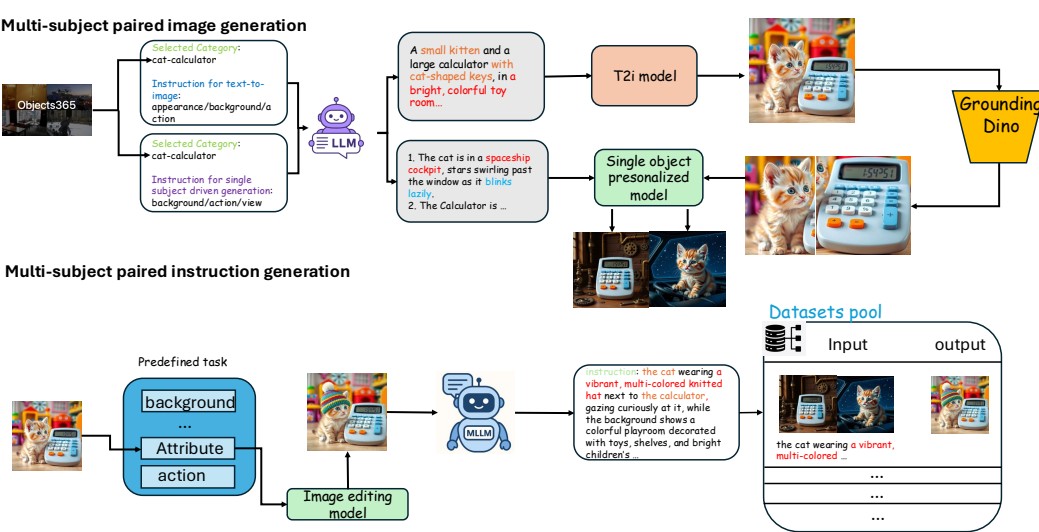

Figure 2: The construction pipeline of our dataset

We attribute these shortcomings to two factors: the lack of high-quality multi-subject personalization datasets and the absence of refined post-training strategies. For multi-subject driven datasets, existing methods such as OmniGen (Xiao et al., 2025) introduce X2I-subject, where most of the high-quality data focuses on human faces, while subject datasets for general scenarios are constructed from GRIT (Peng et al., 2023) and exhibit relatively low consistency. UNO (Wu et al., 2025b), on the other hand, generates subject pairs using a T2I model, which inherently introduces discrepancies in consistency.

To address this, we leverage strong single-subject personalization models such as FLux Kontext to introduce a scalable multi-subject data generation pipeline. Specifically, we generate images containing multiple subjects with a T2I model, apply segmentation, and then perform subject-specific personalization via image editing models. This pipeline yields a large-scale dataset of 350K multi-subject images. Moreover, existing benchmarks for multi-subject personalized generation are neither fine-grained nor comprehensive. For instance, DreamBooth (Ruiz et al., 2023) simply relies on DINO (Liu et al., 2024) similarity between the generated image and the reference subject at the global level, while XVerseBench (Chen et al., 2025) evaluates generated images via detection and segmentation but does not segment the reference subject images, nor does it consider scenarios with duplicated subjects. Therefore, to comprehensively evaluate multi-subject personalization, we propose PSRBench, a new benchmark comprising seven subsets, each assessing models from three perspectives: subject consistency, image aesthetics, and semantic alignment.With these data in place, we first conduct supervised fine-tuning on a single-subject personalization model and introduce a scalable frame-wised positional encoding that equips the model with multi-subject personalization knowledge. This encoding scheme generalizes effectively to varying numbers of input reference images. Moreover, existing methods are limited to the SFT stage, where the optimization objective is defined at the global image level, making it difficult to ensure subject consistency. To address this limitation, we propose the Pairwise Subject Consistency Reward (PSR), which is combined with other semantic alignment rewards to perform reinforcement learning–based fine-tuning of the model.

Extensive experiments demonstrate that our method achieves state-of-the-art performance across multiple subsets, with both quantitative and qualitative results validating its effectiveness as shown in Figure 1 and Figure 5.

Our contributions are summarized as follows:

- We propose a scalable multi-subject personalization data generation pipeline that leverages existing single-subject models to synthesize data with arbitrary numbers of subjects. Using this pipeline, together with predefined data-cleaning strategies, we construct approximately 350K high-quality samples for multi-subject personalized generation.

- We extend the scalable frame-wise positional encoding and introduce Pairwise Subject Consistency Reward, which together, through a two-stage training paradigm, substantially enhance the model's ability to maintain subject consistency and to faithfully adhere to textual instructions.

- We present PSRBench, a fine-grained and comprehensive benchmark with seven subsets, evaluating multi-subject personalization across three dimensions: subject consistency, aesthetic preference, and semantic alignment.

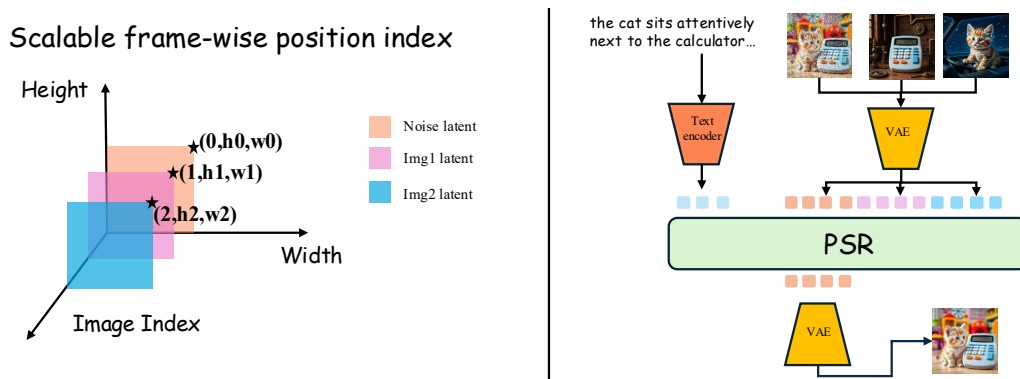

Figure 3: Left: Scalable frame-wise position index. Right: Our SFT strategy, which concatenates tokens from different images along the token dimension.

## 2 RELATED WORKS

### 2.1 MULTI-SUBJECT DRIVEN GENERATION

Personalized generation refers to the task of synthesizing images of a target subject in novel scenes, given reference images containing that subject. Existing approaches to this task can be broadly categorized into two paradigms: test-time fine-tuning and zero-shot methods. Early approaches such as DreamBooth (Ruiz et al., 2023) and Textual Inversion (Gal et al., 2022) adapt a model by fine-tuning selected parameters on 3–5 reference images of the subject, enabling the model to capture the desired concept. However, this paradigm requires training a separate model for each concept, incurring substantial computational and storage costs.

More recently, research has shifted toward zero-shot methods (Labs et al., 2025; Tan et al., 2024; Tao et al., 2025), which aim to perform personalization for arbitrary subjects without the need for model fine-tuning. While such single-subject personalization methods demonstrate promising results, their performance remains limited when multiple subjects are involved. To address the challenges of multi-subject personalization, UNO (Wu et al., 2025b) leverages text-to-image models to construct data and perform co-evolution, first training on single-subject data before extending to multiple subjects. XVerse (Chen et al., 2025) achieves subject control by transforming reference images into shifts for text-stream modulation. OmniGen2 (Wu et al., 2025a) constructs high-quality identity-preserving data from video sources. Despite these advances, existing approaches still struggle to ensure consistency across multiple subjects and are difficult to scale effectively to larger sets of reference images.

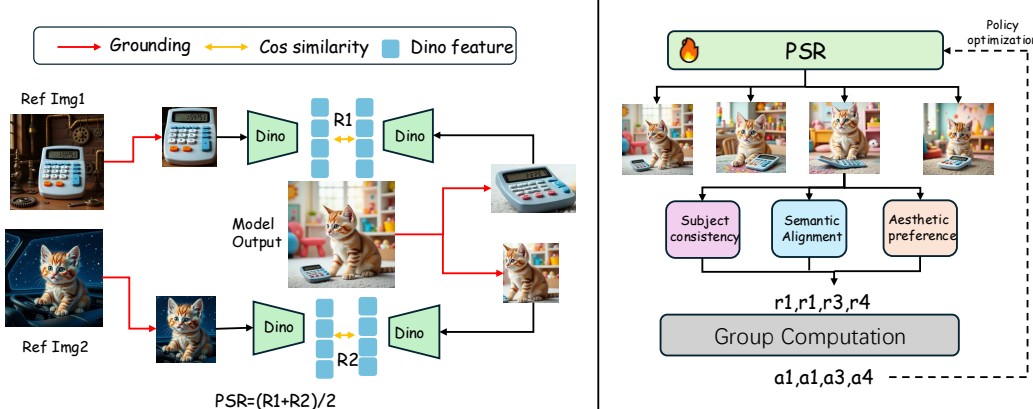

Figure 4: Left: Pairwise subject rewards. Right: GRPO training pipeline combining PSR with multiple rewards.

## 2.2 REINFORCEMENT LEARNING FOR GENERATION

Reinforcement learning (RL) (Guo et al., 2025; Schulman et al., 2017; Ouyang et al., 2022), as a paradigm for alignment, has already demonstrated substantial potential in the field of visual generation (Liang et al., 2024; Jiang et al., 2025; Duan et al., 2025; Xue et al., 2025; Liu et al., 2025).

For autoregressive text-to-image models, T2I-R1 integrates semantic-level and token-level chain-of-thought reasoning and employs diverse rewards to enhance text alignment during generation. In the case of flow-matching–based models, approaches such as Flow-GRPO (Liu et al., 2025) and Dance-GRPO (Xue et al., 2025) leverage different forms of reward to fine-tune text-to-image generation models via RL, thereby significantly improving their capabilities in semantic fidelity, text rendering, and human preference alignment.

However, applying reinforcement learning to multi-subject personalized generation remains an open challenge.

## 3 DATASETS AND BENCHMARK

The core of our dataset construction pipeline lies in leveraging the powerful capabilities of large language models, text-to-image models, and single-image personalization models in a synergistic manner, enabling the creation of datasets that can scale to an arbitrary number of personalized subjects. Specifically, as shown in Figure 2 our dataset construction pipeline consists of two stages: paired image generation and paired instruction generation.

### 3.1 SCALABLE MULTI-SUBJECT DRIVEN DATASETS

**multi-subject paired image generation**  Existing approaches for synthesizing personalized generation data often rely solely on text-to-image (T2I) models combined with strict subject-consistency filtering strategies, such as UNO (Wu et al., 2025b) and Subject200K (Tan et al., 2024), where a T2I model generates a diptych containing the same subject depicted across two different scenes. However, due to the inherent instability of T2I models in producing such data, these methods often yield datasets of limited consistency and quality.

In contrast, recent advances in single-subject personalization models have demonstrated strong capabilities: given a single subject, these models can reliably generate its appearance across novel scenes while maintaining high consistency. Motivated by the capability of these models, we propose a highly scalable multi-subject paired image generation pipeline that harnesses the strengths of these powerful models. Specifically, as shown in Figure 2, for constructing data with $n$ subjects, we first sample an n-element category set $\mathcal{C} = \{c_1, c_2, \ldots, c_n\}$ from the full category pool of Object365 (Shao et al., 2019). We then prompt a large language model (LLM) to generate a text-

to-image instruction $T^{t2i}$, where each category's appearance is explicitly specified to increase data diversity. In addition, we prompt the LLM to produce subject-driven instructions $T^{sub}$ for each category, which will be used for generating single-subject images. Using a state-of-the-art T2I model, we employ $T^{t2i}$ to synthesize multi-subject images $I_{out}$ With the aid of a grounding-based object detection model, we detect instances in $I_{out}$ by category using Grouding Dino (Liu et al., 2024), and crop them via bounding boxes to obtain single-subject images $I_{crop}$. Finally, given $I_{crop}$ and $T^{sub}$, we generate new reference images $I_{ref}$ for each subject, which serve as the input references of our dataset.

**multi-subject paired instruction generation**  In Stage 1, to guide the T2I model toward producing higher-quality and more distinctive images, the prompts include explicit descriptions of the subjects' appearances. However, directly reusing these descriptions for multi-subject personalization may cause the model to exploit textual leakage—focusing only on the appearance information revealed in the text rather than attending to the reference images themselves. To address this issue, we perform recaptioning of $I_{out}$ for tasks such as background and positional relations. Moreover, to support broader application scenarios, we design tasks involving personalized attribute binding and employ editing models to modify $I_{out}$ accordingly to satisfy these task requirements.

The key idea of this stage is to leverage MLLMs to recaption generated images under different tasks, and to employ advanced image editing models to further refine Iout Concretely, for tasks involving multi-subject attribute binding, we first use an LLM to generate task-specific editing instructions. These instructions, together with $I_{out}$, are then fed into an editing model to produce $I_{out}^*$. Next, we apply an MLLM to recaption $I_{out}$, explicitly using pronouns to specify individual subjects while avoiding descriptions of their visual appearances.

This stage yields a large set of high-quality, diverse, and personalized instruction data, which substantially enriches the dataset and supports more effective downstream training.

## 3.2 PSRBench

Existing benchmarks for evaluating multi-subject personalization exhibit significant limitations in both evaluation tasks and metrics. For example, DreamBench only includes combinations of two subjects with overly simplistic scene descriptions. Moreover, current evaluation protocols are often coarse-grained. UNO, for instance, computes the DINO score between each subject in a generated two-subject image and its corresponding reference image, but such a method cannot provide a fine-grained assessment of subject consistency. OmniContext leverages GPT-4.1 for evaluation, yet it still fails to reliably measure consistency. XVerse proposes a segmentation-based approach, where each subject in the generated image is segmented and then compared to the corresponding input reference using DINO scores. However, this method assumes that reference images contain isolated subjects on plain white backgrounds—an idealized setting that does not generalize well to real-world scenarios, where reference images typically include subjects within simple or even complex backgrounds.

To this end, we propose PSRBench, a comprehensive and multi-dimensional benchmark for multi-subject personalization. Specifically, our benchmark consists of seven subsets, each representing a distinct sub-task: attribute, background, action, positional relations, complex prompts, three-subject generation, and four-subject generation. Each subset is evaluated along three complementary dimensions: subject consistency, semantic alignment, and aesthetic preference. For subject consistency, we adopt a grounding-based approach: both input and output images are first processed with an object grounding model to detect and crop the subjects, after which DINO scores are computed on corresponding subject pairs. This design enables precise and fine-grained evaluation of subject consistency. For semantic alignment, we also adopt a grounding-based evaluation to assess positional consistency: the centers of the subjects are detected, and their relative positions are compared against the prompt. For other sub-tasks of semantic alignment, we employ an MLLM (Bai et al., 2025) to evaluate image-text consistency with respect to the specific task requirements. For aesthetic preference, we rely on hpsv3 (Ma et al., 2025) for evaluation.

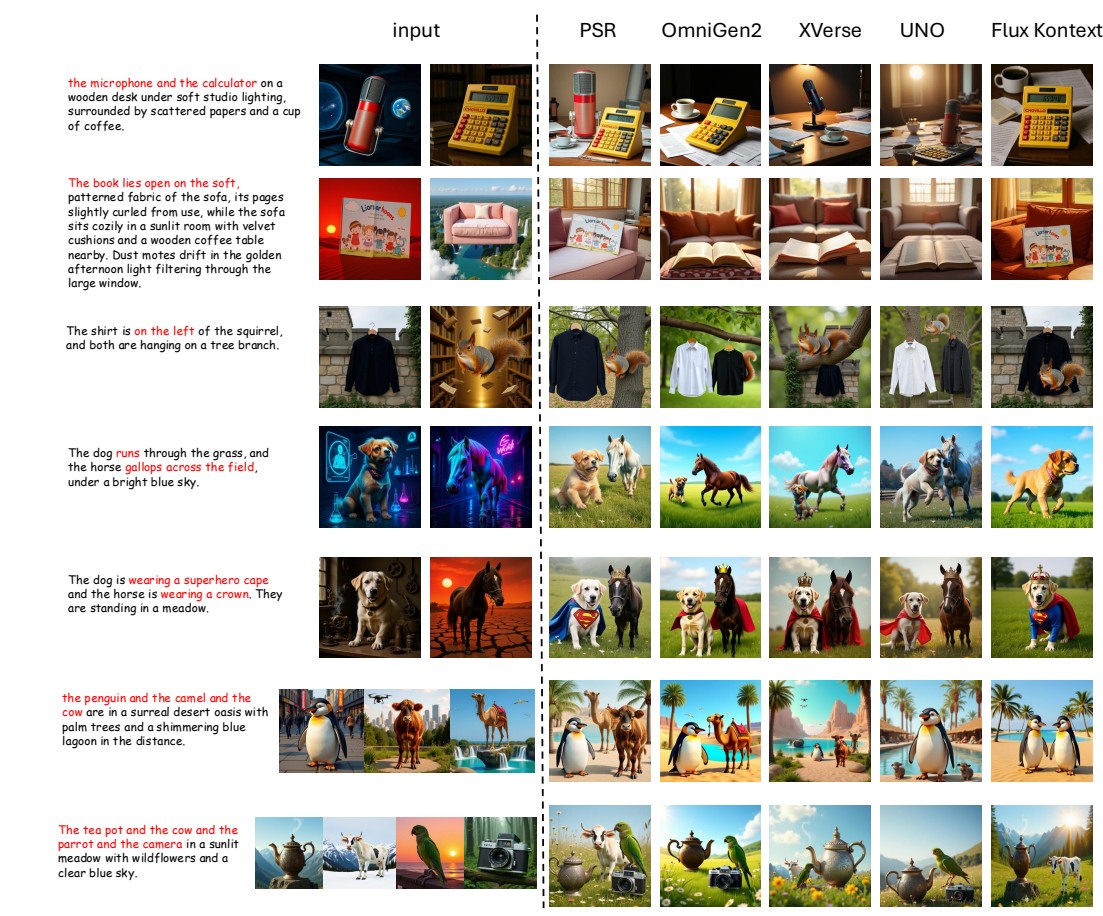

Figure 5: Qualitative analysis results of PSR, where the figure illustrates the performance of different methods on each subset of PSRBench. Our approach achieves the best performance in maintaining subject consistency and semantic alignment.

## 4 METHODS

### 4.1 PRELIMINARY

**Flow Matching**    Flow Matching model is trained by minimizing the objective:

$$\min_v \int_0^1 \mathbb{E}\left[\left|(z_1 - z_0) - v(z_t, t)\right|^2\right] dt \qquad (1)$$

During the sampling process, the model starts from random noise, and the ODE is solved using a simple Euler solver:

$$z_{t_{i-1}} = z_{t_i} + v(z_t, t)\Delta t \qquad (2)$$

**GRPO for flow matching**    Flow-GRPO (Liu et al., 2025) transforms the ODE-based sampling into an SDE form to incorporate the stochasticity required by GRPO. Through Euler-Maruyama discretization, the resulting policy update is as follows:

$$x_{t+\Delta t} = x_t + \left(v_\theta\left(x_t, t\right) + \frac{\sigma_t^2}{2t}\left(x_t + (1-t)v_\theta\left(x_t, t\right)\right)\right)\Delta t + \sigma_t\sqrt{\Delta t}\epsilon, \quad \epsilon \sim \mathcal{N}(0, I) \qquad (3)$$

Flow-GRPO (Liu et al., 2025) set $\sigma_t = a\sqrt{\frac{t}{1-t}}$, $a$ is a scalar hyper-parameter that controls the noise level.

Table 1: Quantitative results of subject consistency. Ours-SFT denotes the results obtained using only the first-stage training.

| Model | attribute | background | action | position | complex | three | four | **Overall** |
|---|---|---|---|---|---|---|---|---|
| Kontext (Labs et al., 2025) | 0.54 | 0.57 | 0.60 | 0.59 | 0.58 | 0.48 | 0.44 | 0.543 |
| UNO (Wu et al., 2025b) | 0.59 | 0.60 | 0.63 | 0.64 | 0.60 | 0.52 | 0.43 | 0.572 |
| Omnigen2 (Wu et al., 2025a) | **0.65** | 0.63 | 0.70 | 0.70 | 0.67 | 0.60 | 0.54 | 0.642 |
| XVerse (Chen et al., 2025) | 0.59 | 0.65 | 0.73 | 0.67 | 0.68 | 0.59 | 0.50 | 0.630 |
| Ours-SFT | 0.62 | 0.68 | 0.69 | 0.67 | 0.68 | 0.62 | 0.59 | 0.650 |
| Ours | **0.65** | **0.71** | **0.75** | **0.74** | **0.73** | **0.66** | **0.61** | **0.693** |

Table 2: Quantitative results of Aesthetic preference.

| Model | attribute | background | action | position | complex | three | four | **Overall** |
|---|---|---|---|---|---|---|---|---|
| Kontext (Labs et al., 2025) | 0.82 | 1.05 | 0.80 | 0.84 | 1.00 | 0.99 | 0.95 | 0.921 |
| UNO (Wu et al., 2025b) | 0.89 | 1.15 | 0.90 | 1.14 | 1.04 | 1.13 | 1.08 | 1.047 |
| Omnigen2 (Wu et al., 2025a) | 0.85 | 1.10 | 0.89 | 1.14 | 1.03 | 1.08 | 1.09 | 1.026 |
| XVerse (Chen et al., 2025) | 0.82 | 0.93 | 0.74 | 0.99 | 0.87 | 0.93 | 0.95 | 0.890 |
| Ours-SFT | 0.69 | 0.96 | 0.72 | 0.92 | 0.87 | 0.99 | 1.06 | 0.887 |
| Ours | **0.92** | **1.23** | **0.97** | **1.24** | **1.08** | **1.29** | **1.31** | **1.149** |

## 4.2 SCALABLE FRAME-WISED POSITIONAL ENCODING

In the SFT stage, our goal is to endow a single-image personalization model with the knowledge required to handle multi-image, multi-subject scenarios. To achieve this, we leverage an frame-wise positional offset together with a multi-image joint training strategy.

Similar to prior work, we also employ a VAE to encode input images, and then concatenate the resulting latents with the noise latent. In the case of multi-image inputs, this token-level concatenation remains a natural choice; however, it requires a specific positional indexing scheme to distinguish tokens from different input images. Some previous approaches introduced offsets along the h and w dimensions—for example, UNO suggested that the spatial index of the $i - th$ image should start from the terminal position of the $(i - 1) - th$ image. While such indexing allows better utilization of pretrained model capacity, it suffers from two key limitations:

By enforcing offsets along the h and w dimensions, the model implicitly inherits a strong prior—that the second image is naturally positioned to the right or below the first image—which complicates fine-grained control through textual prompts.

When scaling to more images, e.g., three or four inputs, such large spatial offsets in h and w hinder the model's generalization ability.

Therefore, as shown in Figure 3, we extend the positional encoding scheme proposed in Flux Kontext by employing only a virtual temporal offset to indicate the index of each input image. Concretely, for the latent tokens of the i-th input image, the positional offset is defined as

$$PO = (i, h, w), \tag{4}$$

where $h, w$ denote the latent's spatial dimensions.

During training, we adopt a multi-image joint training strategy that incorporates datasets with varying numbers of input images. We argue that jointly training across different reference counts allows the model to exploit their complementary benefits, leading to improved generalization and robustness in multi-subject personalization.

## 4.3 PAIRWISE SUBJECT-CONSISTENCY REWARDS

In multi-subject personalization scenarios, the input reference images often contain background elements, whereas our objective is to endow the model with end-to-end capability to handle such realistic inputs.

To this end, we adopt an online reinforcement learning framework to post-train the model initialized from the first-stage SFT. For multi-subject personalization, we introduce a novel Pairwise Subject-Consistency Reward (PSR) as shown in Figure 4. The key idea behind PSR is subject decoupling:

Table 3: Quantitative results of Semantic Alignment.

| Model | attribute | background | action | position | complex | three | four | **Overall** |
|---|---|---|---|---|---|---|---|---|
| Kontext (Labs et al., 2025) | 0.68 | 0.81 | 0.46 | 0.16 | 0.68 | 0.65 | 0.59 | 0.576 |
| UNO (Wu et al., 2025b) | 0.70 | 0.91 | 0.62 | 0.48 | 0.73 | 0.81 | 0.69 | 0.706 |
| Omnigen2 (Wu et al., 2025a) | 0.83 | 0.92 | **0.81** | 0.49 | 0.75 | 0.83 | 0.74 | 0.767 |
| XVerse (Chen et al., 2025) | 0.80 | 0.89 | 0.60 | 0.53 | **0.77** | 0.82 | 0.71 | 0.731 |
| Ours-SFT | 0.84 | 0.92 | 0.66 | 0.51 | 0.69 | 0.85 | 0.84 | 0.759 |
| Ours | **0.85** | **0.93** | 0.63 | **0.54** | 0.69 | **0.89** | **0.87** | **0.771** |

we disentangle each subject from the global image, and then encourage pairwise similarity between the disentangled subjects and their references to guide training.

Formally, let a pretrained model $\theta$ take multiple input images , each containing multiple subjects, and produce an output $I_{out}$. We apply subject decoupling on the output to obtain subject-specific crops: $I_{dec}^i = g(I_{out}, c_i)$, where $g$ denotes an open-vocabulary object detector and $c_i$ represents the category of the $i - th$ subject. The decoupled image $I_{dec}^i$ contains only the subject-specific region. Since input references in real-world scenarios also contain background, we perform the same decoupling operation on input references: $I_{gt}^i = g(I_{ref}^i, c_i)$ We then define the subject consistency reward as the average similarity between each pair of corresponding subject crops:

$$\text{PSR} = \frac{1}{N} \sum_i f\left(I_{\text{dec}}^i, I_{\text{gt}}^i\right) \tag{5}$$

where $f$ measures visual similarity.

To further mitigate the risk of copy-paste artifacts and prevent reward hacking, we incorporate two additional reward signals. The overall reward is defined as follows:

$$R = w1 * PSR + w2 * R_s + w3 * R_h \tag{6}$$

where $R_s$ is semantic reward computed by Qwen-2.5-vl (Bai et al., 2025):

$$R_s = MLLM(instruction, I_{out}), \tag{7}$$

$R_s$ is aesthetic preference reward from a human preference scoring model HPSv3 (Ma et al., 2025):

$$R_h = HPS(instruction, I_{out}), \tag{8}$$

## 5 EXPERIMENTS

### 5.1 IMPLEMENTATION DETAILS

We build our multi-subject personalization model on top of Flux Kontext. In the first stage, we set the learning rate to 1e-4. Our dataset provides 2–4 images as inputs, and during training we sample them with probabilities of 0.9:0.05:0.05, enabling joint training across different reference counts. In the second stage, we use a smaller learning rate of 1e-5. Specifically, we adopt a multi-reward joint training strategy. The reward for subject consistency is provided by our proposed Pairwise Subject-consistency Reward (PSR). For textual faithfulness, we leverage Qwen-2.5-vl-7B to evaluate image-text alignment, while human preference is modeled using hpsv3. Since this stage requires evaluating human preferences, sampling and training are both conducted over the original 28 diffusion timesteps. All square images are resized to a resolution of 512×512, while non-square images are resized to the nearest dimension around 512. Experiments are conducted on 8 NVIDIA A800 GPUs.

We compare our proposed PSR with state-of-the-art multi-subject personalization approaches, including UNO (Wu et al., 2025b), OmniGen2 (Wu et al., 2025a), XVerse Chen et al. (2025), and Flux Kontext Labs et al. (2025). Since Flux Kontext only supports single-image inputs, for fair evaluation we concatenate multiple input images along the width dimension before feeding them into the model.

### 5.2 QUALITATIVE RESULT

As shown in Figure 5, presents seven qualitative analyses on PSRBench. In comparison with other baselines, our approach demonstrates superior performance in both subject preservation and text

Table 4: Comparison of different offset strategies.

| Method | 2 subjects | 3 subjects | 4 subjects |
|---|---|---|---|
| w/ height-width | 0.62 | 0.56 | 0.53 |
| w/ width-offset | 0.65 | 0.59 | 0.55 |
| w/ height-offset | 0.65 | 0.60 | 0.54 |
| w/ frame-wise offset | **0.68** | **0.62** | **0.59** |

adherence. In the case of two-image inputs, our method is able to faithfully retain the original appearance of the subjects while simultaneously complying with semantic instructions, thereby validating the effectiveness of both our data and methodology. For example, our approach successfully preserves the distinctive visual characteristics of the "microphone" and "calculator," while also adhering to the textual description that specifies "scattered papers and a cup of coffee." In contrast, competing methods often fail to maintain high consistency across multiple subjects, with XVerse even suffering from complete subject omission. Furthermore, our results show that the proposed method exhibits strong text alignment capabilities: when the prompt contains positional, action, or attribute-binding constraints, our model can still generate coherent images. It is worth noting that although OmniGen2 also achieves relatively strong performance in text adherence, it underperforms in terms of subject consistency.

When scaling to scenarios involving a larger number of input images, existing methods degrade substantially. As illustrated in Figure 5, with four-subject inputs, competing approaches consistently exhibit subject omission, whereas our method continues to maintain subject consistency. This highlights the strong scalability and robustness of our approach.

### 5.3 QUANTITATIVE RESULT

PSRBench employs three distinct metrics to evaluate subject consistency, semantic alignment, and human preference. Table 1 compares the subject consistency of existing methods. Benefiting from high-quality paired data and reinforcement learning with PSR-based multi-reward optimization, our approach achieves the highest subject consistency score of 0.69, and demonstrates significant superiority over prior methods on both the three-subject and four-subject subsets.

Tables 2 and Tables 3 report the human preference scores and semantic alignment scores, respectively. PSR also achieves the overall best performance on these two metrics, indicating that our method not only preserves consistency in multi-subject scenarios but also exhibits strong text controllability, enabling the generation of high-quality images.

### 5.4 ABLATION STUDY

We further conduct ablation studies on the scalable position index. Table 4 reports the results of training with different positional encodings on subject consistency. The results demonstrate that the scalable position index offers better scalability and performs more effectively in multi-subject scenarios.

## 6 CONCLUSION

In this work, we addressed the challenges of multi-subject personalized image generation, where existing models struggle to maintain subject consistency and adhere to textual instructions. To overcome these limitations, we proposed a scalable multi-subject data generation pipeline and introduced a Pairwise Subject-Consistency Reward within a reinforcement learning framework. Furthermore, we designed PSRBench, a comprehensive benchmark that evaluates subject consistency, semantic alignment, and human preference across diverse and challenging scenarios. Extensive experiments demonstrate that our method not only achieves state-of-the-art performance in multi-subject consistency but also exhibits strong scalability and text controllability, enabling the generation of high-quality and semantically faithful images. We believe this work provides a solid foundation for future research on controllable and scalable personalized generation.

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

# A APPENDIX

## A.1 THE USE OF LARGE LANGUAGE MODELS

The paper only uses LLMs for language polishing and grammar checking.

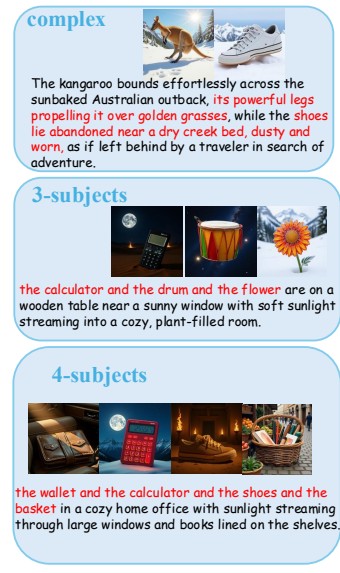
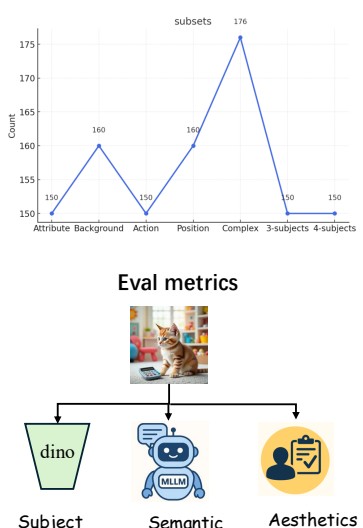

Figure 6: PSRBench

## A.2 FLOW GRPO

GRPO, as a powerful reinforcement learning method, optimizes the policy function through group advantages.

$$\hat{A}_t^i = \frac{R\left(x_0^i, c\right) - \text{mean}\left(\left\{R\left(x_0^j, c\right)\right\}_{j=1}^G\right)}{\text{std}\left(\left\{R\left(x_0^j, c\right)\right\}_{j=1}^G\right)} \tag{9}$$

$x_0^i$ denotes the $i - th$ sample, and $G$ represents the number of groups. Flow-GRPO proposes using GRPO to optimize the sampling process by training the policy through the maximization of a regularized objective.

$$\mathcal{J}_{\text{Flow-GRPO}}\left(\theta\right) = \mathbb{E}_{\boldsymbol{c} \sim \mathcal{C}, \{\boldsymbol{x}^i\}_{i=1}^G \sim \pi_{\theta_{\text{old}}}\left(\cdot|\boldsymbol{c}\right)} f(r, \hat{A}, \theta, \varepsilon, \beta) \tag{10}$$

where,

$$f(r, \hat{A}, \theta, \varepsilon, \beta) = \frac{1}{G}\sum_{i=1}^G \frac{1}{T}\sum_{t=0}^{T-1}\left(\min\left(r_t^i(\theta)\hat{A}_t^i, \text{clip}\left(r_t^i(\theta), 1-\varepsilon, 1+\varepsilon\right)\hat{A}_t^i\right) - \beta D_{\text{KL}}\left(\pi_\theta\|\pi_{\text{ref}}\right)\right) \tag{11}$$

$$r_t^i(\theta) = \frac{p_\theta\left(x_{t-1}^i \mid x_t^i, c\right)}{p_{\theta_{\text{old}}}\left(x_{t-1}^i \mid x_t^i, c\right)} \tag{12}$$

## A.3 PSRBENCH

The analysis of PSRBench is shown in the Figure 6, where a representative case is presented for each subset. For every subset, we conduct evaluations from three complementary dimensions.

