# OpenReview forum: "PSR: Subject-Consistency Rewards for Multi-Subject Personalized Generation"
_ICLR.cc/2026/Conference — ICLR 2026 Conference Withdrawn Submission_

### Official Review · Reviewer_ea43 · 2025-10-27

**Soundness:** 2
**Presentation:** 3
**Contribution:** 4
**Rating:** 4
**Confidence:** 3

**Summary:**

This work addresses the challenge in the current personalized image generation field, where it is difficult to simultaneously ensure textual alignment and maintain the fidelity of reference subjects. To solve this problem, an innovative attribute learning module is introduced. Additionally, a synthetic dataset named CoupleX is constructed, which features subject-paired samples focused on depicting activities and interactions in natural scenes. The authors’ method demonstrates promising results.

**Strengths:**

- This work puts forward a scalable multi-subject personalization data generation pipeline, which is expected to significantly advance the development of this research field.
- The study presents PSRBench, a comprehensive benchmark designed to facilitate standardized evaluation and comparison in related tasks.
- The frame-wise positional encoding and Pairwise Subject Consistency Reward proposed in this work successfully maintain subject consistency while enabling the model to faithfully adhere to textual instructions.

**Weaknesses:**

- Both the frame-wise position index and Pairwise Subject Consistency Reward lack novelty in design and implementation.
- The two claims presented in Lines 356 and 359 are not intuitively obvious. Furthermore, they lack sufficient theoretical justification and experimental validation to support their validity.
-  The design of the ablation experiment is overly simplistic. It fails to demonstrate the necessity of each module and the specific role each component plays in improving the model’s performance.

**Questions:**

Should the text's position index be modified to align with the proposed Scalable frame-wise position index?

---

### Official Review · Reviewer_Qn3q · 2025-10-30

**Soundness:** 3
**Presentation:** 2
**Contribution:** 3
**Rating:** 4
**Confidence:** 4

**Summary:**

The paper tackles the challenge of multi-subject personalized image generation, where existing models struggle to maintain subject consistency and follow textual prompts. It introduces a scalable data generation pipeline that leverages single-subject models to synthesize 350K high-quality multi-subject samples, along with PSRBench, a comprehensive benchmark covering seven subsets and three evaluation dimensions. The core contribution is a two-stage training framework combining supervised fine-tuning with scalable frame-wise positional encoding and reinforcement learning using a novel Pairwise Subject-Consistency Reward (PSR) that improves alignment between generated subjects and references. Extensive experiments show clear performance gains over state-of-the-art baselines in both consistency and semantic adherence, demonstrating the effectiveness and scalability of the proposed approach.

**Strengths:**

1. Novel reward design: The proposed Pairwise Subject-Consistency Reward effectively isolates per-subject consistency signals, overcoming the global-image limitations of prior methods.
2. Benchmark contribution: PSRBench is thorough, multi-dimensional, and addresses clear shortcomings in prior evaluations like DreamBench and XVerseBench.

**Weaknesses:**

1. The paper lacks comparison and discussion with DreamBench++ (Peng et al., 2024), which replaces DINO-based evaluation with GPT-based metrics that better correlate with human evaluation. Since the proposed method also relies on DINO and HPSv3, exploring a GPT-based evaluator could offer a more human-aligned reward and potentially enhance both subject consistency and semantic fidelity.
2. Ablation studies are limited—particularly regarding the contribution of individual reward terms (PSR, semantic, aesthetic).

**Questions:**

1. How does the PSR-based reward balance subject identity preservation versus potential overfitting to reference appearance?
2. Can the authors provide examples where PSR leads to failure cases (e.g., identity drift, blending artifacts)?

---

### Official Review · Reviewer_mFhk · 2025-11-02

**Soundness:** 2
**Presentation:** 2
**Contribution:** 2
**Rating:** 4
**Confidence:** 4

**Summary:**

This paper tackles flaws in multi-subject personalized image generation—existing models often lose subject consistency or ignore text prompts—via three key contributions.

First, it builds a 350K-sample dataset: using LLMs to write prompts, T2I models to generate multi-subject images, object detectors to crop reference subjects, and editing models to refine details, solving data scarcity.

Second, it proposes a two-stage training: SFT with scalable frame-wise positional encoding (tagging subjects to avoid confusion) and RL with Pairwise Subject-Consistency Reward (PSR) (averaging similarity between generated and reference subjects). Additional rewards (semantic alignment via Qwen-2.5-VL, aesthetics via HPSv3) boost performance.

Third, it introduces PSRBench: 7 task subsets (e.g., 4-subject generation) evaluating 3 dimensions (consistency, alignment, aesthetics).
Experiments show it outperforms UNO, OmniGen2, etc., with top scores in all metrics, avoiding subject omission and wrong attribute assignment.

**Strengths:**

1. This paper proposes a multi-subject personalized image generation training dataset.
2. This paper also proposes a benchmark covering 7 subsets and 3 dimensions.  Both the training dataset and the benchmark may be useful for the image generation community.

**Weaknesses:**

1. The description of your reinforcement learning part (GRPO for flow matching) is insufficient, which doesn't propose any insight to explain why you use this training method and why it is useful.
2. The experiment part is also insufficient. You present the results of only your proposed PSRBench, which may be unpersuasive without other common benchmark results.
3.  This paper only conducts ablation studies on the scalable position index. How can you prove the effectiveness of the proposed training dataset and training strategy?

**Questions:**

1. In my opinion, the main contribution of this paper is the proposed training dataset and the benchmark. So will the training dataset and benchmark be released as open-source?
2. The contribution of the proposed training method should be further demonstrated by more experiments on other benchmarks and more detail ablation studies.

---

### Note · Authors · 2025-11-12

I have read and agree with the venue's withdrawal policy on behalf of myself and my co-authors.